

# Coverage and quality of DNA barcode references for Central and Northern European Odonata

Matthias Geiger[1], Stephan Koblmüller[2], Giacomo Assandri[3], Andreas Chovanec[4], Torbjørn Ekrem[6], Iris Fischer[5,7,8], Andrea Galimberti[9], Michał Grabowski[10], Elisabeth Haring[5,7,8], Axel Hausmann[11], Lars Hendrich[11], Stefan Koch[12], Tomasz Mamos[10], Udo Rothe[13], Björn Rulik[1], Tomasz Rewicz[10], Marcia Sittenthaler[7], Elisabeth Stur[6], Grzegorz Tończyk[10], Lukas Zangl[2,14,15] and Jerome Moriniere[16]

[1] Zoologisches Forschungsmuseum Alexander Koenig (ZFMK) - Leibniz Institute for Animal Biodiversity, Bonn, Germany
[2] Institute of Biology, University of Graz, Graz, Steiermark, Austria
[3] Area per l'Avifauna Migratrice, Istituto Superiore per la Protezione e la Ricerca Ambientale (ISPRA), Ozzano Emilia, BO, Italy
[4] Federal Ministry of Agriculture, Regions and Tourism, Vienna, Austria
[5] Department of Functional and Evolutionary Ecology, University of Vienna, Vienna, Austria
[6] Department of Natural History, NTNU University Museum, Norwegian University of Science and Technology, Trondheim, Norway
[7] Central Research Laboratories, Natural History Museum Vienna, Vienna, Austria
[8] Department of Evolutionary Biology, University of Vienna, Vienna, Austria
[9] Department of Biotechnology and Biosciences, ZooPlantLab, University of Milano - Bicocca, Milano, Italy
[10] Department of Invertebrate Zoology and Hydrobiology, University of Łódź, Łódź, Poland
[11] SNSB-Zoologische Staatssammlung, München, BY, Germany
[12] Independent Researcher, Mindelheim, BY, Germany
[13] Naturkundemuseum Potsdam, Potsdam, BB, Germany
[14] ÖKOTEAM - Institute for Animal Ecology and Landscape Planning, Graz, Steiermark, Austria
[15] Universalmuseum Joanneum, Studienzentrum Naturkunde, Graz, Steiermark, Austria
[16] AIM - Advanced Identification Methods GmbH, Leipzig, SN, Germany

Corresponding author
Matthias Geiger,
m.geiger@leibniz-zfmk.de

## ABSTRACT

**Background:** Dragonflies and damselflies (Odonata) are important components in biomonitoring due to their amphibiotic lifecycle and specific habitat requirements. They are charismatic and popular insects, but can be challenging to identify despite large size and often distinct coloration, especially the immature stages. DNA-based assessment tools rely on validated DNA barcode reference libraries evaluated in a supraregional context to minimize taxonomic incongruence and identification mismatches.

**Methods:** This study reports on findings from the analysis of the most comprehensive DNA barcode dataset for Central European Odonata to date, with 103 out of 145 recorded European species included and publicly deposited in the Barcode of Life Data System (BOLD). The complete dataset includes 697 specimens (548 adults, 108 larvae) from 274 localities in 16 countries with a geographic emphasis on Central Europe. We used BOLD to generate sequence divergence metrics and to examine the taxonomic composition of the DNA barcode clusters within the dataset and in comparison with all data on BOLD.

**Results:** Over 88% of the species included can be readily identified using their DNA barcodes and the reference dataset provided. Considering the complete European dataset, unambiguous identification is hampered in 12 species due to weak mitochondrial differentiation and partial haplotype sharing. However, considering the known species distributions only two groups of five species possibly co-occur, leading to an unambiguous identification of more than 95% of the analysed Odonata via DNA barcoding in real applications. The cases of small interspecific genetic distances and the observed deep intraspecific variation in *Cordulia aenea* (Linnaeus, 1758) are discussed in detail and the corresponding taxa in the public reference database are highlighted. They should be considered in future applications of DNA barcoding and metabarcoding and represent interesting evolutionary biological questions, which call for in depth analyses of the involved taxa throughout their distribution ranges.

# INTRODUCTION

Dragonflies (Anisoptera) and damselflies (Zygoptera) of the order Odonata constitute a nearly cosmopolitan group of insects of about 7,000 species (*Kalkman et al., 2008*). The group is recorded from all continents except Antarctica and has its highest levels of diversity in the tropics (*Kalkman et al., 2008*). Their relatively large size, striking morphology, sometimes bright coloration and interesting biology make them popular among nature lovers (*Garrison, von Ellenrieder & Louton, 2006*). Due to the comparatively small number of species occurring in Europe (<150, see *Kalkman et al., 2010*), their life history strategies, ecological requirements and distribution are relatively well-studied. Odonata is therefore also one of the few insect taxa comprehensively covered in national and international Red List assessments. The most recent European Red List considers about 15% of the 138 assessed species as threatened (*Kalkman et al., 2010*). The most important threats to odonates are desiccation of their habitats due to global warming, and intensified water usage for consumption and agriculture (*Kalkman et al., 2010*).

Dragonflies and damselflies are also routinely used for assessment and monitoring of ecological status within the European Water Framework Directive (WFD, Directive 2000/60/EC; *European Commission, 2000*). For instance, a dragonfly association index was developed for Austria in order to assess the ecological status of rivers within the system of the Water Framework Directive and to evaluate restoration measures (*Chovanec & Waringer, 2015*). Despite our wide knowledge of odonates, some species and especially their larvae and exuviae—which are crucial for judging autochthony—cannot be readily and easily identified in the field. This is among the reasons why comparatively few species are used in regular WFD assessments. In Germany, for instance, less than a quarter of the occurring species are used (*Haase, Sundermann & Schindehütte, 2006*).
DNA-based methods for species identification known as DNA barcoding and DNA metabarcoding allow more comprehensive assessments of aquatic communities (*Hajibabaei et al., 2011*; *Taberlet et al., 2012*; *Elbrecht et al., 2017*; *Leese et al., 2016*, *2018*). It has been also demonstrated that non-invasive approaches relying on DNA and cells from filtered water samples can be effective in detecting relevant taxa (e.g., *Hajibabaei et al., 2012*; *Majaneva et al., 2018*; *Zizka et al., 2019*). Crucial for accurate and effective metabarcoding of environmental- or bulk sample DNA is a comprehensive and high quality database of reference DNA barcodes, built on correctly identified, well-documented and vouchered specimens (*Weigand et al., 2019*). A recent analysis on the availability of reference data for DNA-based assessment of European aquatic taxa showed that dragonfly species are among the best covered insect taxa (*Weigand et al., 2019*). Yet, of the 1206 DNA barcodes *Weigand et al. (2019)* analyzed only 251 representing 49 species were then public and could be quality-checked. Ten percent of the species with DNA barcodes were represented by one specimen only and 6% of the species were missing from the DNA barcode reference library completely (June 12$^{th}$, 2019). When the more reliable threshold of five DNA barcodes per species was assessed, the proportion of available sequences dropped to ca. 60% of the species with less than 20% of them being publicly available. The remaining sequences and associated metadata were at the time private in the Barcode of Life Data Systems (BOLD, www.boldsystems.org; *Ratnasingham & Hebert, 2007*) and could therefore not be evaluated. Regional representation is shown to be important for accurate identification using DNA Barcoding (e.g., *Bergsten et al., 2012*). A wider geographic coverage was lacking for many odonate species when *Weigand et al. (2019)* assessed the order (e.g., no public data from Italy, only 12 species from Germany, 13 from Poland, 2 from Norway, etc.), thereby increasing the probability of underestimating molecular diversity (e.g., from refugial areas; *Galimberti et al., 2021*) and also hampering routine DNA-based assessments by lacking reliable reference barcodes. The latest significant additions to the reference library for European Odonata in BOLD were those from Malta and Italy (*Rewicz et al., 2021*; *Galimberti et al., 2021*). The latter publication presented the first comprehensive DNA barcode library for Italian odonates and examined diversity and distribution of mitochondrial lineages on the Holarctic scale.

The present study brings together results from different European DNA barcoding initiatives, compiles the available data for joint evaluation, and presents new data to facilitate DNA-based monitoring approaches of aquatic ecosystems. Feeding high quality data into public repositories such as BOLD is not trivial since thorough evaluation of the data before publication is needed to avoid and reduce noise stemming from mis- or un-identified specimens (*Trebitz et al., 2015*; *Rulik et al., 2017*; *Collins & Cruickshank, 2013*; *Becker, Hanner & Steinke, 2011*). Iterative processes for the validation of generated DNA barcode data should ideally incorporate both new and published data. The process of assembling near-complete reference libraries for species-rich taxa bears the chance to also identify taxonomic inconsistencies during the process, and the possibility to discuss and correct them before deposition of premature taxonomic hypotheses. Often, those issues can only be addressed through comparisons of material collected throughout a

species' distribution range, including material of closely related species. Such data also makes it possible to investigate if DNA barcoding accuracy is influenced by spatial effects as documented for beetles (*Bergsten et al., 2012*), moths (*Hausmann et al., 2013*) and fish (*Geiger et al., 2014*).

The "BIN Discordance" analysis in BOLD is a tool for the validation of newly generated data that helps to reveal two different cases of potential taxonomic inconsistencies: (a) specimens assigned to different species that share a molecular operational taxonomic unit (termed BIN—barcode index number), or (b) specimens of a particular species are assigned to two or more BINs. For the majority of studied insect orders, a BIN very often represents a close species-proxy as delineated by traditional taxonomy (e.g., for Lepidoptera, *Hausmann et al., 2013*). However, some genera or families throughout all taxa exhibit problems with simple species delineation based on DNA barcodes. This is often due to high intra- or low inter-specific genetic distances (see *Hubert & Hanner, 2015*). Every "disagreement/conflict" case is then the starting point for re-evaluation of both DNA sequences and morphological data. We follow the concept of Integrative Taxonomy (*Padial et al., 2010*; *Schlick-Steiner et al., 2010*; *Fujita et al., 2012*) to infer whether there are potential, previously overlooked species in our data. This is a widely recognized phenomenon when analysing comprehensive DNA barcode datasets and can impact not only macroecology and conservation issues but also reveal fascinating research objects pointing at evolutionary mechanisms of cryptic speciation (*Struck & Cerca, 2020*).

Several such cases have been published where DNA sequences have helped revealing hidden species level diversity in odonates (*Mitchell & Samways, 2005*; *Damm, Schierwater & Hadrys, 2010*; *López-Estrada et al., 2020*; *Vega-Sánchez, Mendoza-Cuenca & Gonzalez-Rodriguez, 2020*). Neglecting the presence of cryptic diversity does not only lead to underestimation of factual biodiversity, but also can impair indicator species approaches as different taxa react differently to stressors (*Zettler et al., 2013*; *Macher et al., 2016*). But also the opposite—low or absent mitochondrial divergence between species—is expected to occur in large DNA barcode datasets. While this can prevent ready delineation of species with one mitochondrial marker only, it is important to document these cases as starting points for further in-depth studies or revisions (e.g., "warnings" as in *Galimberti et al., 2021*).

In this publication we analyze available and newly generated DNA barcodes for European damselflies and dragonflies to (i) evaluate quality of the new data through comparisons with available DNA barcodes in BOLD, (ii) reveal cases of shared BINs between species indicative for the need of more thorough studies, and (iii) detect potential cases of cryptic diversity, i.e., when the same morphological species is assigned to two or more BINs.

## MATERIALS & METHODS

A network of over 30 institutional taxonomists and external specialists collected and contributed specimens to the different DNA barcoding campaigns, in number of specimens primarily from Germany (SNSB-ZSM, ZFMK), Poland (University of Lodz), Italy (University of Milano-Bicocca, private collection G. Assandri) and Norway (through

the NTNU University Museum and Norwegian Barcode of Life Network, NorBOL). Additional specimens were contributed from Austria (NHMW, University of Graz) and Romania (Babes-Bolyai University).

## Field permits

For samples from Germany field work permits were issued by the responsible state environmental offices in Bavaria [Bayerisches Staatsministerium für Umwelt und Gesundheit, for the project: "Barcoding Fauna Bavarica"] and from the Amt für Natur- und Landschaftsschutz, Rhein-Sieg-Kreis (67.1–1.03–19/2016KRO). Italian specimens were collected in part in protected areas and some of the collected species are included in the EU Habitats Directive. The Italian Ministry of the Environment, Land and Sea released a national permit for the collection of species included in European and Italian conservation directives or to collect samples in regional or national protected areas (Prot. n° 0031783.20-11-2019). Austrian specimens were collected with permits from the provincial governments of Burgenland (A4/NN.AB-10097-5-2017 and A4/NN.AB-10200-5-2019), Lower Austria (RU5-BE-1489/001-208; RU5-BE-64/018-2018), Styria (ABT13-53S-7/1996-156 and ABT13-53W-50/2018-2) and Vienna (MA22-169437/2017). Polish specimens from Wigierski National Park were collected under the permit no. 12/2018 issued by the Park authorities to Grzegorz Tończyk. All the other material did not require additional permits for legal collection. None of the collected specimens from Norway were from areas where sampling is restricted. Thus, sample permits were not required.

## Material studied

Most studied specimens were adults (see below) of which the majority were stored in >96% EtOH prior to DNA extraction. The remaining samples were derived from dry-preserved material. Specimen ages at the time of sequencing ranged from 0–4 years (79%) to up to 16 years (21%). For subsequent analyses we aimed to select only DNA barcode sequences >500 bp, which fulfilled the requirements for being assigned to a BIN. However, in order to achieve maximum taxonomic coverage and to fill gaps in the catalogue of Central European Odonata through combined evaluation of DNA barcodes we included 39 partial DNA barcodes shorter than 500 bp and 9 sequences publicly available in BOLD (8 mined from GenBank). The number of specimens available per species ranged from 1 (15 singletons) to 49 in *Ischnura elegans* (Vander Linden, 1820) (mean = 6.8; SD = 7.4). For problematic taxa, i.e., species that shared BINs or species that were assigned to more than one BIN and whose patterns could not be explained by our data, we downloaded additional DNA barcode sequences available on BOLD with the hope that data covering a larger geographic distribution will allow us to elucidate the process underlying BIN sharing or deep intraspecific divergence. Detailed information on collection sites and dates is available in Supplemental File S1. Voucher information such as life stage, locality data, habitat, altitude, collector, identifier, taxonomic classifications, habitus images, DNA barcode sequences, primer pairs, and trace files are publicly accessible in the "DS-ODOGER—DNA barcode references for Central and Northern European Odonata" dataset in BOLD (http://www.boldsystems.org–https://doi.org/10.5883/DS-ODOGER).
The respective voucher specimens are deposited in public and private collections listed in Supplemental File S1.

## Laboratory protocols

Protocols for generating DNA barcodes varied between laboratories and projects due to either being organism specific or taxonomically broadly oriented.

For samples analysed at the CCDB, submitted by SNSB-ZSM, NorBOL, DNAqua-Net from Poland: A tissue sample was removed from each specimen and transferred into 96 well plates for subsequent DNA extraction, PCR and bi-directional Sanger sequencing at CCDB. All protocols for DNA extraction, PCR amplifications, and Sanger sequencing procedures are available online under: ccdb.ca/resources/. Samples were first amplified with a cocktail of standard and modified Folmer PCR primers CLepFolF (5′–ATT CAA CCA ATC ATA AAG ATA TTG G) and CLepFolR (5′–TAA ACT TCT GGA TGT CCA AAA AAT CA) targeting the full DNA barcode fragment (see *Hernández-Triana et al., 2014*). The same primers were employed for subsequent bidirectional Sanger sequencing reactions (see also *Ivanova et al., 2007*). A second PCR round was conducted for a selection of samples that did not amplify in the first attempt and targeted the mini-barcode (313 bp) proposed by *Leray et al. (2013)* with primers dgHCO2198 and mlCOIintF.

The samples submitted to CCDB through DNAqua-Net from Poland were amplified with the primers OdoF1_t1 (5′–TGT AAA ACG ACG GCC AGT ATT CAA CHA ATC ATA ARG ATA TTG G) and OdoR1_t1 (5′–CAG GAA ACA GCT ATG ACT AAA CTT CTG GAT GYC CRA ARA AYC A) and subsequently sequenced with M13 primers.

ZFMK: Tissue sub-sampling, DNA extraction, polymerase chain reaction (in house at ZFMK) and sequencing (BGI Genomics) followed standard protocols (*Astrin et al., 2016*) and are described in detail in *Rulik et al. (2017)*.

KFUG: DNA extraction followed a rapid Chelex protocol described in *Richlen & Barber (2005)*, subsequent PCR, clean-up and bidirectional Sanger sequencing were performed following *Duftner, Koblmüller & Sturmbauer (2005)* and *Koblmüller et al. (2011)* using the primers ODO_LCO1490d (5′-TTT CTA CWA ACC AYA AAG ATA TTG G) and ODO_HCO2198d (5′-TAA ACT TCW GGR TGT CCA AAR AAT CA) (*Dijkstra et al., 2014*).

NHMW: DNA extraction was performed with the DNeasy Blood and Tissue Kit (Qiagen) using the standard protocol specified by the company. For PCR amplification two primer sets were used (*Haring et al., 2020*): the complete sequence of the cytochrome c oxidase subunit 1 gene (COI) plus partial sections of the flanking tRNA genes were amplified with the primers Tyr-Odo-F (5′-CTC CTA TAT AGA TTT ACA GTC T-3′) and Leu-Odo-R (5′-CTT AAA TCC ATT GCA CTT TTC TGC C-3′) resulting in amplicon lengths of ~1,660 bp (54 °C annealing temperature). Alternatively, primers CO1-Odo-F5 (5′-TGC GAC RA TGR CTG TTT TC-3′) and CO1-Odo-R6 (5′-TGC ACT TTT CTG CCA CAT TAA A-3′) were combined to amplify almost the complete COI gene (amplicon length 1,532 bp, 47 °C annealing temperature). PCR was performed in a volume of 50 µl containing 0.5 µl Qiagen Taq polymerase (5 units/µl), 5 µl 10 × PCR Buffer, 10 µl Q-Solution (Qiagen, Hilden, Germany), 1.5 mM MgCl2, 2.5 mM dNTP Mix, 0.5 µM of

each primer, and 1 µl DNA template. The PCR cycling protocol included an initial denaturation at 94 °C for 3 min, followed by 35 cycles of denaturation at 94 °C for 1 min, annealing for 30 s, extension at 72 °C for 30 s. The final step was an extension at 72 °C for 10 min. Samples were sequenced bidirectionally using the PCR and two internal primers (CO1-Odo-F3 5′-GAT TCT TTG GAC AYC CHG AAG-3′ and CO1-Odo-R3 5′-GTT TCC TTT TTA CCT CTT TCT TG-3′).

Methods for generating DNA barcodes from additional material from Poland (Lodz) follows *Rewicz et al. (2021)*; the procedures for Italian specimens are described in *Galimberti et al. (2021)*. An overview of the primer combinations used and references thereof are given in the Supplemental File S4.

### Data analysis

Sequence divergences for the COI-5P barcode region (mean and maximum intraspecific variation and minimum genetic distance to the nearest-neighbor species) were calculated using the "Distance Summary" and "Barcode Gap Analysis" tools in BOLD, employing the Kimura-2-Parameter (K2P) distance metric (*Puillandre et al., 2012*) after aligning all sequences >500 bp length with the amino acid HMM based algorithm and pairwise deletion of positions with missing data. BOLD's BIN Discordance analysis was applied to detect either BINs containing different species, or species split into two or more BINs. BOLD groups all DNA barcodes (public and non-public) into clusters of highly similar sequences, which are then assigned unique BIN identifier (*Ratnasingham & Hebert, 2013*). It enables identification of specimens also when taxonomic information is widely lacking (e.g., *Morinière et al., 2019*). As the BIN system is dynamic and dependent on the underlying data, the composition of a BIN can change over time. The analyses in this study were conducted on May18th in 2020.

The mitochondrial relationships based on the DNA barcode region were visualized via Maximum Likelihood (ML) trees inferred in IQ-TREE (*Nguyen et al., 2015*) using the PhyloSuite platform (*Zhang et al., 2020*). Analyses were done separately for Anisoptera and Zygoptera for enhanced presentability and due to presumably different rates of molecular evolution (e.g., *Koroiva & Kvist, 2018*). Best fitting models of evolution were inferred based on the Bayesian Information Criterion (BIC) in ModelFinder (*Kalyaanamoorthy et al., 2017*). These models—GTR+F+R5 and GTR+F+R3 for the Anisoptera and Zygoptera datasets, respectively—were then applied for ML tree searches with 10,000 ultrafast bootstrap replicates conducted in IQ-TREE. Separate ML trees were inferred in IQ-TREE for representative examples for (i) taxa that shared BINs (*Anax* spp. with model HKY+F+I) or (ii) showed deep intraspecific divergence (*Cordulia aenea* with model TIM2+F+G4). For cases of deep intraspecific divergence, additional sequences of closely related taxa not present in our initial dataset were downloaded from BOLD and included in the analysis.

## RESULTS

### Reference library & dataset description

Success in DNA barcode creation ranged from 58% (SNSB and ZFMK—standard protocols at CCDB and in GBOL) to 100% (Austrian and Italian specimens—Odonata-tailored
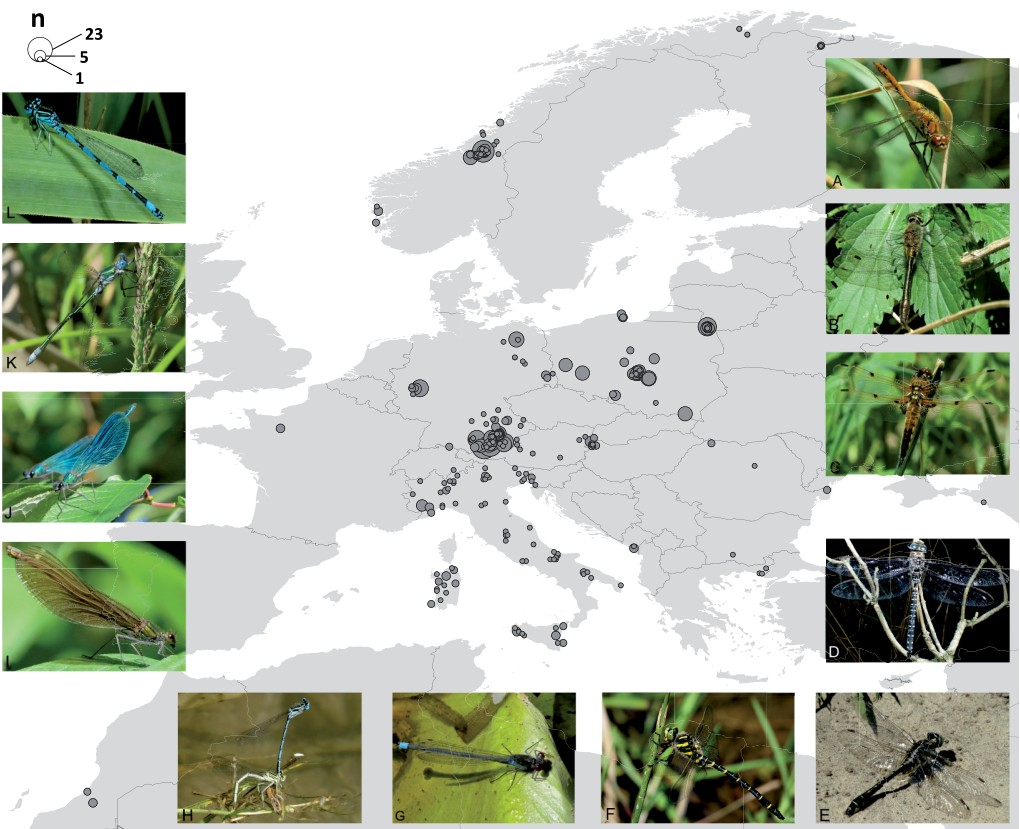

**Figure 1 Map of the 274 sampling locations for the 663 specimens with GPS coordinates deposited in BOLD and pictures for a representative for each family.** Circle size corresponds to the number of specimens analysed from the respective site. Pictures show representatives for each family: ANISOPTERA—Libellulidae: *Sympetrum sanguineum* (A), Corduliidae: *Cordulia aenea* (B), Libellulidae: *Libellula quadrimaculata* (C), Aeshnidae: *Aeshna subarctica* (D), Gomphidae: *Gomphus vulgatissimus* (E), Cordulegastridae: *Cordulegaster boltonii* (F). ZYGOPTERA—Coenagrionidae: *Erythromma najas* (G), Platycnemididae: *Platycnemis pennipes* (H), Calopterygidae: *Calopteryx splendens*, female (I), *Calopteryx splendens*, male (J), Lestidae: *Lestes sponsa* (K), Coenagrionidae: *Coenagrion mercuriale* (L). Photos: Falk Petzold (F) and Stefan Koch (all other).

protocols). The complete dataset (online in BOLD as DS-ODOGER) contained 697 specimens belonging to 103 species bearing a binomen with a COI sequence length range of 309–692 (mean 631 SD = 73.9) bp. Of those, only 3 (0.43%) have not been assigned a BIN due to insufficient length (<500 bp) and some ambiguities. Thirty-nine records (5.6%) were included with a partial DNA barcode only (<500 bp), most of them (35) resulting from the 2nd PCR approach with the *Leray et al. (2013)* mini-barcode primer pair of SNSB-ZSM. Out of all entries analysed, 663 (95.28%) are documented in BOLD with GPS coordinates for 274 localities (partly approximated due to conservation issues), 656 (94.25%) contain information on life stage (548 adults, 108 larvae). The final dataset had a geographic emphasis on Central Europe (Fig. 1) and contained specimens from Germany (288), Poland (159), Italy (100), Norway (92) and Austria (28). The remaining specimens were collected in Bulgaria (1), France (5), Greece (3), Hungary (2), Montenegro (4), Morocco (5), Netherlands (1), Romania (5), Russia (1), Spain (1) and Sweden (2).

**Table 1 Result of the internal BIN discordance report (BOLD v4) for the 7 BINs of 183 individuals with conflicting species-level information associated when compared to all data in the dataset DS-ODOGER.**

| BIN | BIN URI | conflicting species: number of individuals in DS-ODOGER | comment | records in BIN* |
|---|---|---|---|---|
| BOLD:AAJ0782 | http://boldsystems.org/index.php/Public_BarcodeCluster?clusteruri=BOLD:AAJ0782 | *Coenagrion pulchellum: 27*<br>*Coenagrion puella: 26*<br>*Coenagrion ornatum: 5* | species are known to co-occur | 156 |
| BOLD:ADC4648 | http://boldsystems.org/index.php/Public_BarcodeCluster?clusteruri=BOLD:ADC4648 | *Calopteryx xanthostoma: 2*<br>*Calopteryx splendens: 16* | species are largely allopatric with a small area of co-occurrence | 27 |
| BOLD:ABW6681 | http://boldsystems.org/index.php/Public_BarcodeCluster?clusteruri=BOLD:ABW6681 | *Somatochlora meridionalis: 9*<br>*Somatochlora metallica: 5* | species are largely allopatric | 17 |
| BOLD:ADR7794 | http://boldsystems.org/index.php/Public_BarcodeCluster?clusteruri=BOLD:ADR7794 | *Chalcolestes viridis: 1*<br>*Chalcolestes parvidens: 5* | species are largely allopatric with a small area of co-occurrence | 7 |
| BOLD:ABX6596 | http://boldsystems.org/index.php/Public_BarcodeCluster?clusteruri=BOLD:ABX6596 | *Anax parthenope: 4*<br>*Anax imperator: 12* | species are known to co-occur | 124 |
| BOLD:AAE5570 | http://boldsystems.org/index.php/Public_BarcodeCluster?clusteruri=BOLD:AAE5570 | *Ischnura elegans: 49*<br>*Ischnura saharensis: 2*<br>*Ischnura genei: 3* | species are largely allopatric | 143 |
| BOLD:AAN0925 | http://boldsystems.org/index.php/Public_BarcodeCluster?clusteruri=BOLD:AAN0925 | *Gomphus schneiderii: 2*<br>*Gomphus vulgatissimus:13* | species are largely allopatric | 20 |

**Note:**
* As of 2020-05-18 in BOLD including all records (public and non-public); additional taxa might be included, which are not listed here.

## Data evaluation

The internal BIN discordance report (using BOLD v4 on May 18, 2020) evaluated the resulting 96 BINs with 694 individuals and revealed the presence of conflicting taxonomic assignments in 7 BINs encompassing 183 individuals (Table 1). The remaining 74 BINs (plus 15 singleton BINs) did not contain conflicting taxonomic assignments (496 individuals). Of the 103 taxa assigned to a Linnean binomen based on morphological identification, 86.40% (74 + 15 singletons) are unambiguously discriminated by their DNA barcodes at a European scale.

Extending the BIN discordance evaluation to all other data in the reference library (using BOLD v3) revealed a higher proportion of conflicting signal in the global database: 31 of the 96 BINs (358 records) contained records of mixed taxonomic annotations (6 on genus and 25 on species level). While the 6 genus level conflicts were most likely due to coarse misidentification, sample mix-up in a laboratory, sample number mix-up of specimens in BOLD or nomenclatural changes not applied to all affected datasets in BOLD (e.g., *Lestes* Leach, 1815 and *Chalcolestes* Kennedy, 1920), the remaining 25 BINs with mixed species annotations contained 18 cases listed in Table 2 (in addition to the 7 BINs listed in Table 1).

Of the 103 taxa assigned a Linnean binomen based on morphological identification, at least 77.66% (65 + 15 singletons) are unambiguously discriminated by their DNA barcodes (83.49% when neglecting the genus level discordance).

**Table 2 Result of the global BIN discordance report (BOLD v3) for the 18 BINs with conflicting species-level information associated when compared to all data in BOLD.**

| BIN | BIN URI | conflicting species: number of individuals in DS-ODOGER | comment | records in BIN* |
|---|---|---|---|---|
| BOLD:AAA2218 | http://boldsystems.org/index.php/ Public_BarcodeCluster? clusteruri=BOLD:AAA2218 | *Enallagma cyathigerum*[69], *Enallagma hageni*[28], *Enallagma boreale*[22], *Enallagma annexum*[19], *Enallagma ebrium*[18], *Enallagma divagans*[4], *Enallagma clausum*[3], *Enallagma circulatum*[3], *Enallagma laterale*[2], *Enallagma minusculum*[2], *Enallagma geminatum*[2], *Enallagma carunculatum*[2], *Enallagma sp. DNAS-283-223485*[1], *Enallagma aspersum*[1], *Enallagma sp.*[1] | *E. cyathigerum* is the only *Enallagma* species with a Palearctic distribution, the other twelve species have been described later and are recorded from the Nearctic only. | 694 |
| BOLD:AAA6531 | http://boldsystems.org/index.php/ Public_BarcodeCluster? clusteruri=BOLD:AAA6531 | *Aeshna septentrionalis*[26], *Aeshna caerulea*[7], *Aeshna sitchensis*[3] | | 37 |
| BOLD:AAB2237 | http://boldsystems.org/index.php/ Public_BarcodeCluster? clusteruri=BOLD:AAB2237 | *Sympetrum sanguineum*[20], *Sympetrum striolatum*[1] | identification or sequence submission error very likely | 20 |
| BOLD:AAD5734 | http://boldsystems.org/index.php/ Public_BarcodeCluster? clusteruri=BOLD:AAD5734 | *Pyrrhosoma nymphula*[40], *Pyrrhosoma elisabethae*[5] | | 45 |
| BOLD:AAJ5773 | http://boldsystems.org/index.php/ Public_BarcodeCluster? clusteruri=BOLD:AAJ5773 | *Cordulegaster boltonii*[37], *Cordulegaster trinacriae*[2] | | 39 |
| BOLD:AAJ5811 | http://boldsystems.org/index.php/ Public_BarcodeCluster? clusteruri=BOLD:AAJ5811 | *Aeshna grandis*[20], *Aeshna juncea*[1] | identification or sequence submission error very likely | 21 |
| BOLD:AAK5996 | http://boldsystems.org/index.php/ Public_BarcodeCluster? clusteruri=BOLD:AAK5996 | *Orthetrum cancellatum*[27], *Orthetrum albistylum*[1] | identification or sequence submission error very likely | 26 |
| BOLD:AAK5997 | http://boldsystems.org/index.php/ Public_BarcodeCluster? clusteruri=BOLD:AAK5997 | *Orthetrum brunneum*[19], *Orthetrum lineostigma*[2], *Orthetrum anceps*[1] | identification or sequence submission error very likely | 21 |
| BOLD:ABA9336 | http://boldsystems.org/index.php/ Public_BarcodeCluster? clusteruri=BOLD:ABA9336 | *Orthetrum chrysostigma*[24], *Orthetrum julia*[1], *Orthetrum brachiale*[1] | | 28 |
| BOLD:ABA9406 | http://boldsystems.org/index.php/ Public_BarcodeCluster? clusteruri=BOLD:ABA9406 | *Brachythemis leucosticta*[31], *Brachythemis impartita*[8] | | 40 |
| BOLD:ABW0140 | http://boldsystems.org/index.php/ Public_BarcodeCluster? clusteruri=BOLD:ABW0140 | *Paragomphus genei*[20], *Paragomphus elpidius*[1] | | 23 |
| BOLD:ACG0515 | http://boldsystems.org/index.php/ Public_BarcodeCluster? clusteruri=BOLD:ACG0515 | *Platycnemis pennipes*[26], *Platycnemis latipes*[2][1] | identification or sequence submission error very likely | 28 |
| BOLD:ACI1053 | http://boldsystems.org/index.php/ Public_BarcodeCluster? clusteruri=BOLD:ACI1053 | *Aeshna cyanea*[25], *Aeshna grandis*[1] | | 29 |

| BIN | BIN URI | conflicting species: number of individuals in DS-ODOGER | comment | records in BIN[*] |
|---|---|---|---|---|
| BOLD:ACP4984 | http://boldsystems.org/index.php/Public_BarcodeCluster?clusteruri=BOLD:ACP4984 | *Lestes sponsa[25], Lestes sp.[1]* | | 18 |
| BOLD:ACQ1493 | http://boldsystems.org/index.php/Public_BarcodeCluster?clusteruri=BOLD:ACQ1493 | *Sympetrum depressiusculum[22], Sympetrum frequens[10]* | | 3 |
| BOLD:ACQ2278 | http://boldsystems.org/index.php/Public_BarcodeCluster?clusteruri=BOLD:ACQ2278 | *Cordulegaster trinacriae[14], Cordulegaster boltonii[2]* | | 16 |
| BOLD:ACQ4354 | http://boldsystems.org/index.php/Public_BarcodeCluster?clusteruri=BOLD:ACQ4354 | *Cordulegaster picta[2], Cordulegaster insignis[1]* | | 1 |
| BOLD:ADC1709 | http://boldsystems.org/index.php/Public_BarcodeCluster?clusteruri=BOLD:ADC1709 | *Leucorrhinia rubicunda[5], Leucorrhinia intermedia[1]* | | 5 |

**Note:**
[*] As of 2020-05-18 in BOLD including all records (public and non-public).
[1] Number of individuals with and without COI sequences available.

Of the 25 BINs with mixed species annotations, 14 species contained more than 3 specimens in our dataset and could be tested for the presence of diagnostic nucleotide positions allowing for an unambiguous species assignment with a character-based delineation approach. Five of these 14 species possess at least 1 diagnostic position and are all well resolved in the phylogenetic tree reconstructions (Figs. 2 and 3): *Brachythemis impartita* (Karsch, 1890), *Orthetrum cancellatum* (Linnaeus, 1758), *Platycnemis pennipes* (Pallas, 1771), *Pyrrhosoma nymphula* (Sulzer, 1776), *Sympetrum striolatum* (Charpentier, 1840).

Molecular divergence estimates based on all records with a COI sequence length >500 bp (640 specimens, 87 species) and the K2P-model revealed a mean minimum intraspecific divergence of 0.39% (0–9.45%; 0.79 SD; Supplemental File S2). Inspection of the pairwise distances showed that the extreme maximum value is due to one *Chalcolestes viridis* Vander Linden, 1825 (ZPLOD168-20), which is placed in the *C. parvidens* (Artobolevskii, 1929) cluster in the ML-tree (Fig. 2; see discussion for further details). Excluding this specimen from the pairwise distance calculations resulted in a mean minimum intraspecific divergence of 0.37% (0–4.69%; 0.67 SD).

The mean minimum K2P-distance to each nearest neighbor (NN) species (excluding specimen ZPLOD168-20) was 8.69% (0–17.9%; 5.03 SD) and thus on average 23.5 times the mean intraspecific divergence indicating the presence of a DNA barcode gap between the majority of the studied species (NN distance >0 in 95 species; Supplemental File S2). This estimate, however, might be inflated by the 15 singleton species and only 20 species with 10 or more individuals (although 54 species were represented by individuals from 2 or more countries).

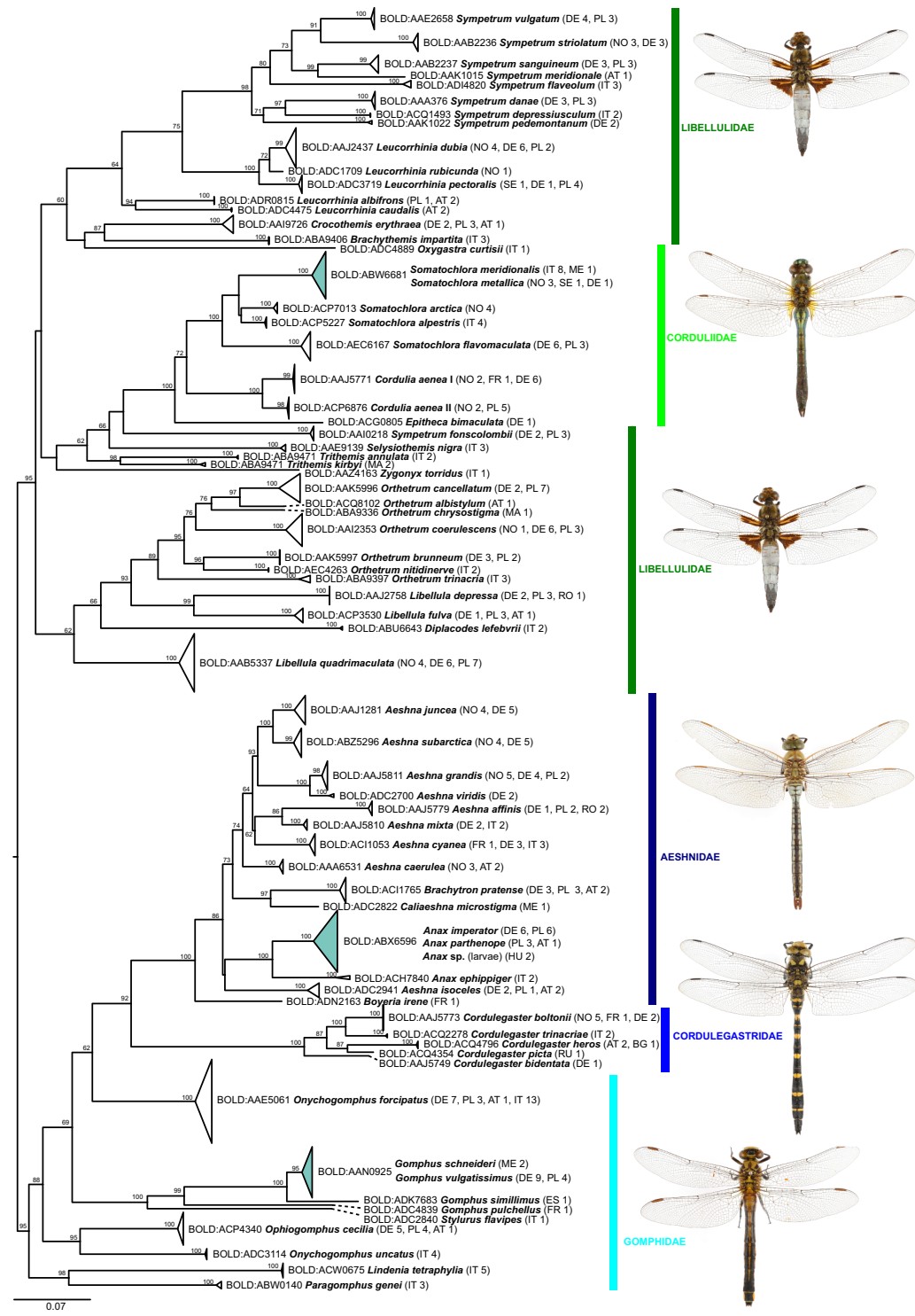

**Figure 2 Maximum likelihood estimation of the phylogenetic relationships for Anisoptera (dragonflies) based on the mitochondrial COI DNA barcode region (GTR+F+R5 model with 10,000 ultrafast bootstrap replicates).**

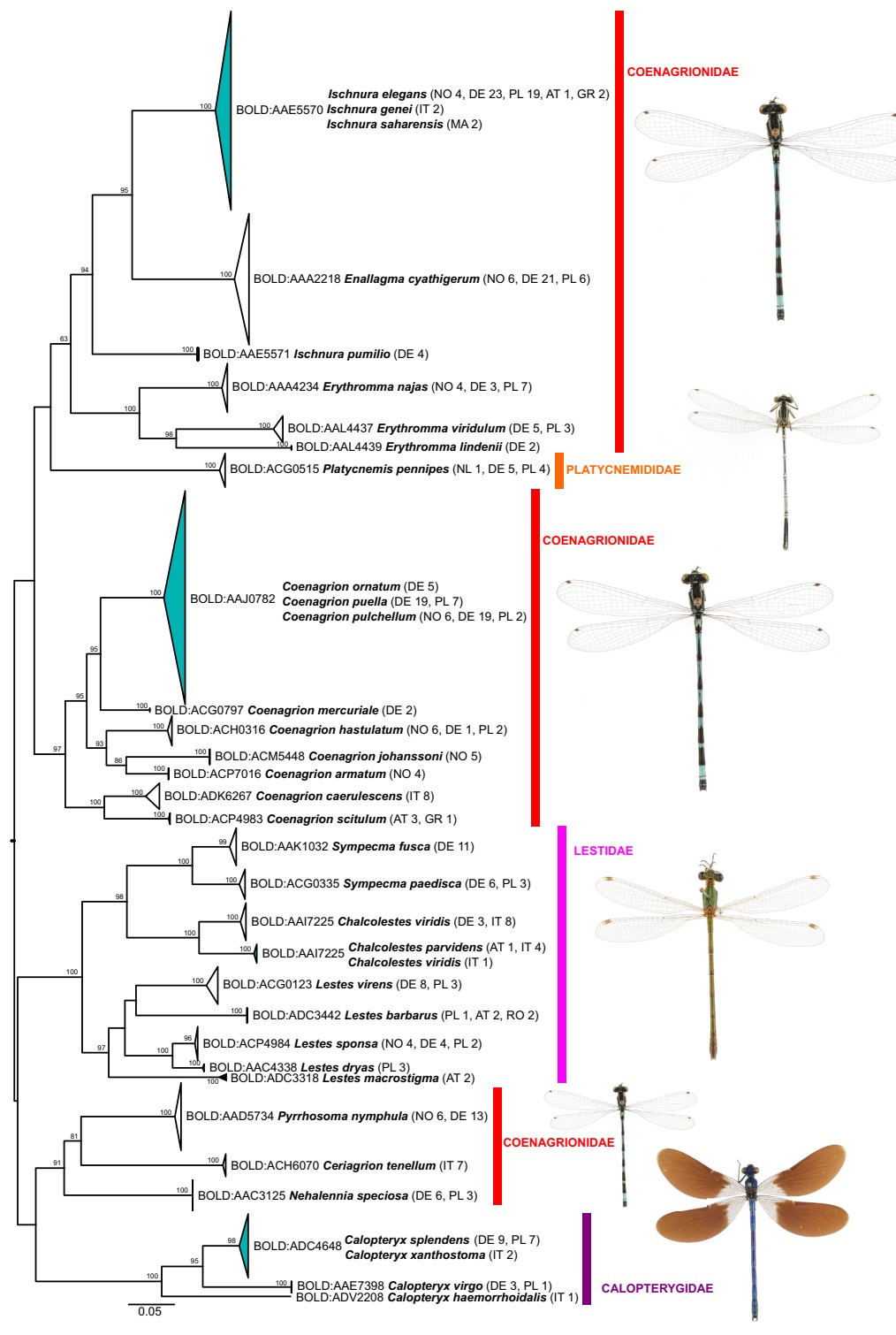

**Figure 3 Maximum likelihood estimation of the phylogenetic relationships for Zygoptera (damselflies) based on the mitochondrial COI DNA barcode region (GTR+F+R3 model with 10,000 ultrafast bootstrap replicates).**

Within genus level sequence divergence was expectedly smaller (9.44–18.34%) than within family level (17.21–25.39%).

The highest intraspecific variation was observed in *Cordulia aenea* (4.69; see also Supplemental File S2), which was also assigned to two different BINs (Fig. 1). Elevated levels of intraspecific variation were also evident in *Onychogomphus forcipatus* (Linnaeus, 1758) (2.97%), but all 17 specimens from 4 countries were, however, united in one common BIN (BOLD:AAE5061).

While the ML bootstrap support for most genus and species level nodes was generally high (>95), family-level and inter-generic relationships were only poorly or not at all supported (Figs. 1 & 2). The latter was more pronounced in dragonflies, but also in damselflies several families were not resolved as monophyletic units.

## DISCUSSION

DNA barcoding is a well-established and powerful tool for quickly gaining preliminary information on the taxonomic status of certain taxa and for assigning problematic specimens, unidentifiable life stages or sexes, or tissue samples to particular species (*Hajibabaei et al., 2007*; *Valentini, Pompanon & Taberlet, 2008*; *Hubert & Hanner, 2015*). This, however, only works when a comprehensive reference database covering the relevant taxa is available and accessible.

The comparison of the success rates for the generation of DNA barcodes for Odonata shows that the standard protocols with highly universal PCR primers (LCO1490 + HCO2198 or LCO1490-JJ + HCO2198-JJ) performed worse than the approaches with Odonata specific primer combinations (Tyr-Odo-F + Leu-Odo-R, CO1-Odo-F5 + CO1-Odo-R6 or ODO_LCO1490d + ODO_HCO2198d). While this is important for targeted, efficient filling of taxonomical gaps with new COI data, it has also implications for routine molecular identification projects where the sub-optimal primer binding might lead to false negative results in a given sample. Yet, the finding that all 39 records with partial DNA barcodes (29 <400 bp; 10 <500 bp) have been assigned successfully to a conspecific BIN cluster (Supplemental File S3) demonstrates that 'minibarcodes'—more suitable for metabarcoding-based community assessments—have the potential to correctly identify Odonata DNA traces. This finding is in line with several recent studies employing partial COI-based DNA metabarcoding for freshwater community assessments (e.g., *Sun et al., 2019*; *Zizka, Geiger & Leese, 2020*).

With the publication of this study together with the examined DNA barcodes we increase the percentage of European dragonflies covered in the reference library to 71% (103 of 145 species; *Boudot & Kalkman, 2015*; with addition in: *Viganò, Janni & Corso, 2017*; *López-Estrada et al., 2020*) considering a threshold of one specimen per species. 88.3% of the 103 species included herein could be readily identified at a European scale, using their DNA barcodes and the reference dataset provided. The rate is even higher if instances of very obvious mis-identifications or mis-labelled sequence depositions in BOLD or GenBank are removed (see Table 2). Because most cases of BIN sharing involve allopatric or parapatric species (Table 1), the re-identification success in the field or at

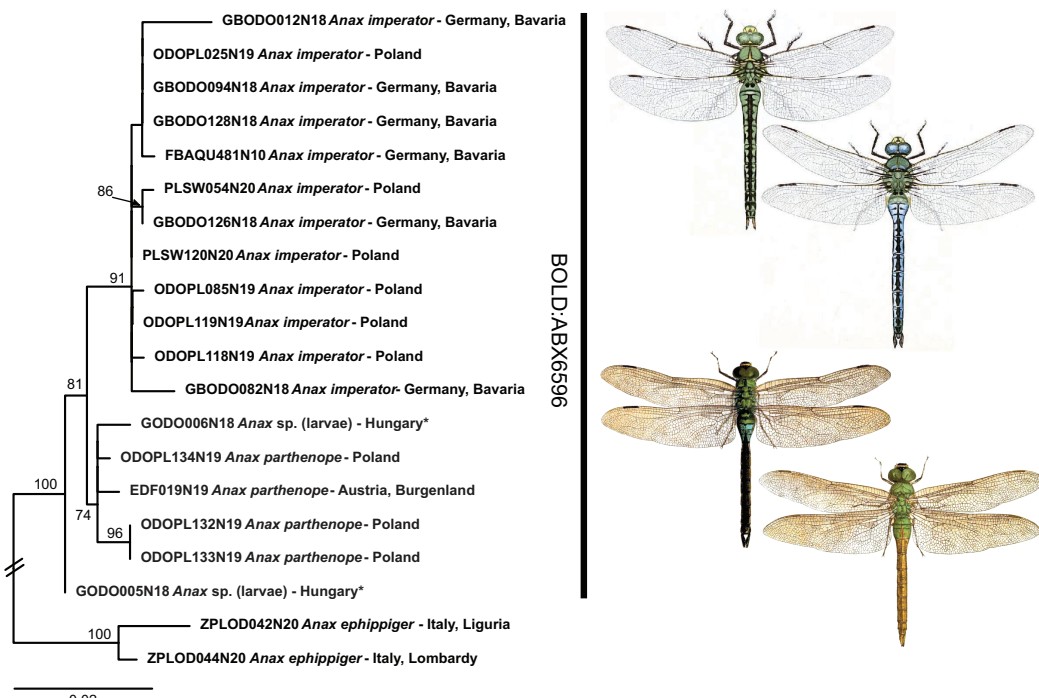

**Figure 4 Example of the occurrence of BIN sharing due to low but consistent mitochondrial differentiation in *Anax imperator* and *A. parthenope*.** Public domain illustrations taken from *Lucas, 1900* (made available from http://www.animalbase.uni-goettingen.de) and the Japanese journal Dobutsugaku zasshi 1901–1903 (made available from the Biodiversity Heritage Library. Contributed by the American Museum of Natural History Library | www.biodiversitylibrary.org). ML-tree inferred with model HKY+F+I.

national and regional scales is certainly higher. In fact, only two groups with five species involved broadly co-occur naturally in Europe: *Anax imperator* Leach, 1815 and *A. parthenope* (Selys, 1839) and three *Coenagrion* species (i.e., *C. pulchellum* (Vander Linden, 1825), *C. puella* (Linnaeus, 1758), and *C. ornatum* (Selys, 1850)).

While shallow genetic distances and sporadic haplotype sharing prevented the unambiguous identification in 12 species taking into account the full dataset (5 of 7 BINs in Table 1), low but consistent mitochondrial differentiation allowed for an unambiguous identification in the remaining two BINs (*Anax imperator*/*A. parthenope* and *Gomphus schneiderii* Selys, 1850/*G. vulgatissimus* (Linnaeus, 1758); Fig. 4 and Supplemental File S3).

The observed identification success rate is comparable to findings from other DNA barcoding studies on dragonflies and damselflies (95% in a set of 51 species from Europe and Africa, *Bergmann et al., 2013*; 79–94%, depending on the criteria used, in a set of 38 species from Brazil, *Koroiva et al., 2017*; 89% in 38 species from the Philippines, *Casas et al., 2018*; 85% in a set of 88 species from Italy, *Galimberti et al., 2021*; 80% of ten species from Malta, *Rewicz et al., 2021*). The rate is also similar to other insect taxa such as Coleoptera (92%; *Hendrich et al., 2015*), Neuroptera (90%; *Morinière et al., 2014*), Heteroptera (92%; *Raupach et al., 2014*) or tachinid flies (93%; *Pohjoismäki, Kahanpää & Mutanen, 2016*), considerably better than for Caelifera (59%; *Hawlitschek et al., 2017*), but

not as good as for e.g., Lepidoptera (99%; *Huemer et al., 2014*), apoid wasps (99%; *Schmid-Egger et al., 2019*), or Ensifera (100%; *Hawlitschek et al., 2017*).

Thus, except for a few cases, DNA barcodes can discriminate between dragon- and damselfly species, making identification through DNA barcoding trustworthy. The relationships between higher taxonomic ranks, such as families and sometimes even genera, on the other hand, were often not resolved as phylogenetically meaningful monophyla (Figs. 1 & 2; Supplemental File S3), with nearest neighbors in the inferred ML-topologies sometimes belonging to taxonomically very distant groups. This, however, is not unexpected, considering that the high substitution rate of mitochondrial genes, which allows for discrimination among species, implies an increased probability of loss of informative characters (saturation) at deeper nodes (e.g., *Morgan, Creevey & O'Connell, 2014*). Thus, inferring robust phylogenetic hypotheses among species and genera must involve sequencing of a much higher number of mitochondrial and/or independently evolving nuclear genetic markers. However, this also means that reliable identification to genus level for species not present in the reference library is difficult for some groups (*Ekrem, Willassen & Stur, 2007*).

Since our dataset includes only a fraction of the entire distribution ranges of many dragonfly and damselfly species, the estimates of intraspecific divergence are not representative for the entire diversity within each species. Consequently, the inferred DNA barcode gaps are very likely an overestimate. Nonetheless, considering the relatively high between-species differentiation observed (mean 8.69% K2P) we are confident that this will also hold true when DNA barcodes of a geographically broader set of individuals are available for all species.

## BIN sharing among species

Cases of BIN sharing (i.e., a lack of a DNA barcode gap) were rare (Tables 1 & 2; Supplemental File S2) and concerned three species triplets and five species pairs. Full barcode/haplotype sharing (distance to nearest neighbor = 0) was observed only among (i) *Calopteryx splendens* (Harris, 1782) and *C. xanthostoma* Charpentier, 1825, (ii) *Coenagrion ornatum*, *C. puella* and *C. pulchellum*, and (iii) *Ischnura saharensis* Aguesse, 1958 and *I. elegans*. However, only in species belonging to *Coenagrion* this might hamper an identification via COI data at certain locations of co-occurrence, as the other two genera should not occur sympatrically except in Liguria (North-West Italy) and France (*Boudot & Kalkman, 2015*). In general, cases of BIN sharing among species might be explained by mitochondrial introgression following hybridization, recent divergence with or without incomplete lineage sorting (ILS) or inadequate taxonomy and misidentification (*Kerr et al., 2007*; *Ward, Hanner & Hebert, 2009*; *Zangl et al., 2020*). The latter can probably be excluded as a source for the observed cases of BIN and haplotype sharing in our study, as most central and northern European Odonata species are easy to identify based on morphological characters and coloration of adults. This is also true for the species that share BINs. Recently, progress has been made in screening odonates for endosymbionts such as *Wolbachia* or *Rickettsia*, for example in *Coenagrion* from the UK (*Thongprem et al., 2020*), or from systems outside of our study area (*Lorenzo-*

*Carballa et al., 2019*—Fiji archipelago; *Salunkhe et al., 2015*–Central India), documenting that these endosymbionts could constitute another route for mitochondrial introgression between species.

In *Calopteryx splendens* and *C. xanthostoma*, BIN sharing might be due to recent divergence and/or hybridization. The two species are evidently closely related (*Weekers, De Jonckheere & Dumont, 2001*). They have largely non-overlapping distributions, but there is evidence for hybridization in regions where they co-occur (*Dumont, Mertens & De Coster, 1993*). This seems to also occur sporadically among more distantly related taxa. For example, *C. haemorrhoidalis* (Vander Linden, 1825) and *C. splendens* (*Lorenzo-Carballa, Watts & Cordero-Rivera, 2014*) and *C. splendens* and *C. virgo* (Linnaeus, 1758) hybridize in at least parts of the region studied herein (*Tynkkynen et al., 2008*; *Keränen et al., 2013*).

The genus *Chalcolestes* includes two BINs, one comprised of *C. viridis*, the other containing *C. parvidens* and one individual of *C. viridis* (also see *Galimberti et al., 2021*). These results are probably due to introgressive hybridization (Supplemental File S3). The two species are morphologically very similar, but do differ in a few characters. Intermediate morphotypes have been reported, mainly from regions where the distribution ranges overlap (*Olias et al., 2007*). Yet, a previous study looking at morphological and genetic differentiation between *C. parvidens* and *C. viridis* in southeastern Europe did not find evidence for hybridization between the two species, suggesting that the intermediate morphotypes are intraspecific variation (*Gyulavária et al., 2011*). As *Gyulavária et al. (2011)* used only a handful of specimens for molecular genetic analyses, the apparent lack of hybridization evidence is not surprising. Where the two species co-occur, there appears to be some prezygotic isolation by temporal segregation in their daily reproductive activities (*Dell'Anna et al., 1996*).

Within *Coenagrion*, previous studies have already reported mitochondrial haplotype sharing between *C. puella* and *C. pulchellum* in England and as morphological characters were sometimes shared between the two species, the observed haplotype sharing was initially attributed to hybridization (*Freeland & Conrad, 2002*). Subsequent genetic analyses based on nuclear microsatellite markers, however, found no evidence for hybridization between these two species (*Lowe et al., 2008*). Whether this pattern is true for the species' entire distribution range remains to be seen, but BIN sharing and haplotype sharing between *C. puella*, *C. pulchellum* and *C. ornatum* across large geographic regions— England, Germany, Norway (*Freeland & Conrad, 2002*, this study)–argue against localized hybridization/introgression events and might indicate either very recent species divergence with ILS or rapid, potentially range-wide, mitochondrial replacement following fairly recent introgression, which appears to be more common in animals than previously thought (*Nevado et al., 2009*; *Good et al., 2015*; *Koblmüller et al., 2017*). Our geographically limited sampling does not permit us to infer the direction of potential introgression with confidence, but the higher genetic diversity in *C. pulchellum* might be an indication that it is the donor species (Supplemental Files S2 and S3). To distinguish between the two alternative scenarios (ILS and mitochondrial replacement), a geographically more comprehensive sample and nuclear multilocus sequence data is

required. Recently, in a preliminary analysis, *Galimberti et al. (2021*, Appendix S7 therein*)* found the three species to be well separated at three nuclear loci.

BIN- and haplotype sharing among species of *Ischnura* (*I. elegans*, *I. genei* (Rambur, 1842), *I. saharensis*) was to be expected (see also *Rewicz et al., 2021*). Together with *I. graellsii* (Rambur, 1842) (not included in our dataset), these species constitute a probably recent radiation around the western Mediterranean basin, with *I. elegans* distributed mostly northeast of the Pyrenees across large parts of Europe, *I. graellsii* south of the Pyrenees to the Atlas, *I. saharensis* southeast of that, and *I. genei* on the Tyrrhenian islands (*Dijkstra & Kalkman, 2012*). In the few regions where two of the species meet, hybridization has been reported (e.g., *Monetti, Sánchez-Guillén & Cordero-Rivera, 2002*; *Sánchez-Guillén et al., 2011*, *2014*; *Wellenreuther et al., 2018*). Hence, both recent divergence and introgression might underlie the observed BIN sharing in this species group.

In dragonflies a very shallow divergence with BIN (and even haplotype) sharing has been recently reported for *A. imperator* and *A. parthenope* (*Galimberti et al., 2021*; *Rewicz et al., 2021*). We also find that these species share one BIN, but unlike the two previously mentioned studies that focused on southern Europe, we do not find evidence for haplotype sharing in our Central European data (Fig. 4). However, our sampling is far from being exhaustive and it might well be that hybridization/introgression is also common in Central European *Anax* Leach, 1815. Hitherto, the two species have only been included in a single phylogenetic analysis based on a single nuclear marker (*Letsch et al., 2009*), and divergence between them was much deeper than between other aeshnid species that are resolved as distinct BINs in our (and other) DNA barcoding datasets. This deeper divergence in the nuclear data would suggest that the observed BIN sharing of the two *Anax* species is indeed due to mitochondrial introgression. However, to conclusively test whether the observed patterns of shallow mitochondrial divergence are due to recent origin or hybridization remains to be studied by means of nuclear multilocus or genome scale data.

BIN sharing was also observed between *Gomphus schneiderii* and *G. vulgatissimus* (Fig. 2 and Supplemental File S3). In Europe, *G. schneiderii* is restricted to the Balkans, whereas *G. vulgatissimus* is found across large parts of Europe, overlapping with *G. schneiderii* in some Balkan regions (*De Knijf, Vanappelghem & Demolder, 2013*). So far, there is no evidence for hybridization between these two species. Even though they share a BIN, low but consistent mitochondrial differentiation allows unambiguous identification.

BIN sharing was also found among *Somatochlora meridionalis* Nielsen, 1935 and *S. metallica* (Vander Linden, 1825) (Fig. 2 and Supplemental File S3; also see *Galimberti et al., 2021*). Although *S. meridionalis* has its main area of distribution in the Balkans and Italy and *S. metallica* is mainly found from Central and northern Europe to Siberia, their distribution areas do overlap in Central Europe and the Balkans. The two species are morphologically very similar, yet show small but consistent differences in larval morphology and adult coloration (*Seidenbusch, 1996*; *Boudot & Kalkman, 2015*). This has previously led to the suggestion that they should indeed be regarded as distinct species despite BIN sharing and the presence of intermediate phenotypes in regions of sympatric

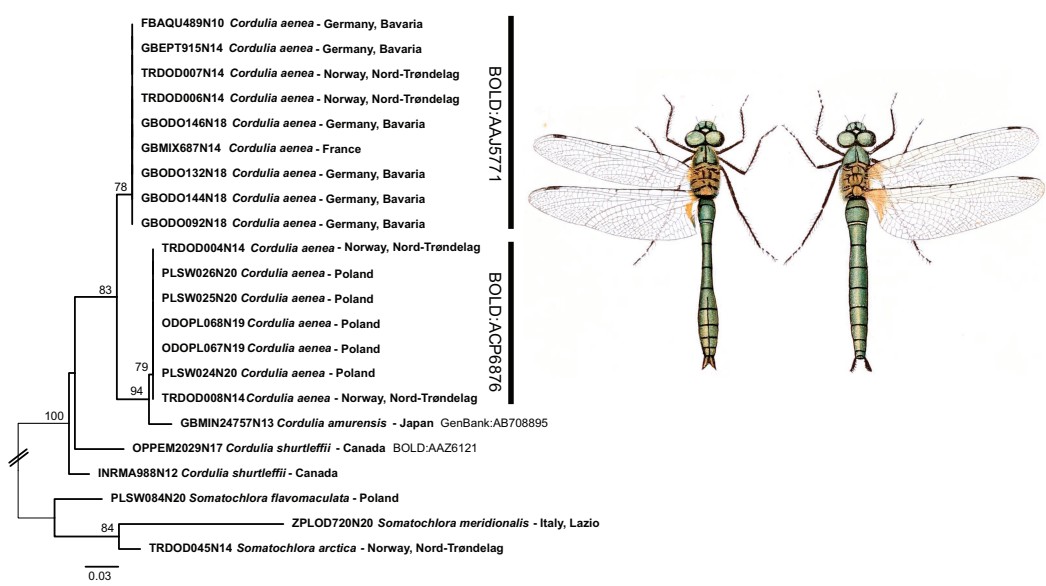

**Figure 5 Example of the occurrence of deep mitochondrial divergence in *Cordulia aenea*.** Public domain illustration taken from *Lucas, 1900* (made available from http://www.animalbase.uni-goettingen.de; male left). ML-tree inferred with model TIM2+F+G4.

occurrence (*Fleck, Grand & Boudot, 2007*). Yet, nuclear multilocus data are required to resolve the phylogenetic relationships and the extent of inter-specific gene flow in this species pair.

Problems with morphological species identification might arise in European regions not covered by our study, especially in southern Europe. In these regions, several morphologically similar species co-occur, some of which are included in our dataset, that are not only more difficult to identify but might also hybridize (e.g., *Ischnura elegans* x *I. graellsii*, *Sánchez-Guillén, Van Gossum & Cordero-Rivera, 2005*; *Sánchez-Guillén et al., 2011*; *Calopteryx* spp., *Dumont, Mertens & De Coster, 1993*; *Weekers, De Jonckheere & Dumont, 2001*; *Lorenzo-Carballa, Watts & Cordero-Rivera, 2014*; *Cordulegaster trinacriae* Waterston, 1976 x *C. boltonii* (Donovan, 1807), *Solano et al., 2018*).

## Deep intraspecific divergence

Whereas the aforementioned species shared DNA barcodes, only one species, *Cordulia aenea*, showed considerable intraspecific sequence divergence. For this species, two deeply divergent clades (up to 4.7% inter-clade K2P divergence) were discovered. One of these two clades includes specimens from Norway, France and Germany, whereas the other one contains specimens from Norway and Poland. Hence, these two clades do not reflect a clear geographic separation. Interestingly, though, this second *C. aenea* clade groups with *C. amurensis* Selys, 1887 from Japan (Fig. 5). Currently, *Cordulia* is considered to comprise three species, *C. aenea*, *C. amurensis* and *C. shurtleffii* Scudder, 1866. In western Eurasia, only *C. aenea* has been reported thus far, with *C. amurensis* restricted to far eastern Asia and *C. shurtleffii* present in North America. Throughout their distribution range, *Cordulia* spp. show quite some variation, but no clear inter-specific differences in

morphology, behavior or ecology are known (*Dijkstra & Kalkman, 2012*), apart from a slightly smaller size of *C. amurensis* as compared to *C. aenea* (*Jödicke, Langhoff & Misof, 2004*) which questioned the validity of the three allopatrically distributed species (*Kosterin & Zaika, 2010*). However, deeply divergent, geographically restricted nuclear genetic lineages were found supporting the existence of three distinct *Cordulia* species in western Eurasia, far eastern Asia and North America, respectively (*Jödicke, Langhoff & Misof, 2004*).

Similar cases of surprisingly high intraspecific divergence were recently observed by *Galimberti et al. (2021)* for different European populations of *Erythromma lindenii* (Selys, 1840) (up to 4.8% inter-clade K2P divergence) and *Coenagrion mercuriale* (Charpentier, 1840) populations highly divergent from other European and North African populations (up to 8% K2P distance). Whether these cases will lead to the formal recognition of species ranks for neglected subspecies need further studies.

## CONCLUSIONS

In this study, we provide the first comprehensive reference DNA barcode collection for central and northern European dragon- and damselflies. We found only five instances of BIN sharing preventing a ready identification of the involved 12 species and one case of deep intraspecific divergence (two BINs for a single species). This implies that the vast majority of central and northern European dragon- and damselflies can be readily identified based on DNA barcodes. While adults may be easily identified by morphological characteristics and color patterns, identification of larvae is often not trivial. As odonates are considered good indicators of environmental health and flagship species for other threatened taxa, and are being used to prioritize areas for conservation action (*Lemelin, 2007*; *Simaika & Samways, 2009*; *Clausnitzer et al., 2017*), reliable species identification is pivotal. The reference DNA barcode data now serve as a basis for a rapid and reliable identification of material/life stages hitherto difficult to identify and community assessments via environmental metabarcoding. The cases with BIN sharing should be considered in applications of DNA barcoding and metabarcoding. These, as well as those with deep intraspecific divergence represent interesting evolutionary biological questions and call for an in depth analysis throughout the species' distribution ranges by means of nuclear multilocus or genomic data.

## ABBREVIATION

| | |
|---|---|
| **CCDB** | Canadian Center for DNA Barcoding |
| **DNAqua-Net** | EU COST Action CA15219 on "Developing new genetic tools for bioassessment of aquatic ecosystems in Europe" |
| **GBOL** | German Barcode of Life initiative |
| **KFUG** | Karl-Franzens-Universität Graz |
| **NHMW** | Naturhistorisches Museum Wien (Natural History Museum Vienna) |
| **SNSB-ZSM** | Staatliche Naturwissenschaftliche Sammlungen Bayerns, Zoologische Staatssammlung München |

| NTNU | Norwegian University of Science and Technology |
| ZFMK | Zoologisches Forschungsmuseum Alexander Koenig, Leibniz Institute for Animal Biodiversity |

## ACKNOWLEDGEMENTS

This study is a result of several long lasting, comprehensive DNA Barcode of Life campaigns (ABOL - Austria; BFB - Bavaria; GBOL - Germany; NorBOL - Norway; PolBOL - Poland), which were only possible with the aid of numerous taxonomic experts and behind-the-scene working logistic helpers. At ZFMK we are indebted to Jana Thormann, Laura von der Mark, Simone Behrens-Chapuis, Moritz Fahldieck, Morris Flecks, Thierry Sellmeier, Wolfgang Wägele and Bernhard Misof. We thank Frank Petzold, Stefan V. Ober and Michael Franzen for contributing to the national DNA barcoding campaigns with their expertise. Jon Kristian Skei, Dag Dolmen, and Steffen Roth are thanked for contributions in sampling and with their expertise in identification of specimens. We are grateful to the DNAqua-Net initiator Florian Leese and the research team at BIO and CCDB in Guelph (Ontario, Canada) for their great support and help and particularly to Sujeevan Ratnasingham for developing the BOLD database (BOLD) infrastructure and the BIN management tools.

### Funding

The GBOL project is funded by grants from the German Federal Ministry of Education and Research (FKZ 01LI1101 and 01LI1501). At ZSM the project was funded by grants from the Bavarian State Ministry of Education and Culture, Science and the Arts (Barcoding Fauna Bavarica, BFB) and the German Federal Ministry of Education and Research (GBOL2: BMBF #01LI1101B). Financial support for the barcoding of Austrian dragon- and damselflies was provided by the Austrian Federal Ministry of Science, Research and Economy in the frame of an ABOL associated project within the framework of the "Hochschulraum-Strukturmittel" Funds, as well as the City of Vienna (Municipal Department 22 - Environmental Protection (MA 22)) and the European Agricultural Fund for Rural Development 2014-2020. Molecular work at the University of Lodz was supported by the Polish NCN Project No 2018/31/B/NZ8/03103 and by the university statutory funds and done during a program 'Capable Pupil Great Student' by Klaudia Nowak in which high school kids may participate in scientific projects at the University of Lodz and by Sylwia Woźniak. Tomasz Rewicz and Tomasz Mamos were supported by ITC Conference Grants, COST Action CA15219-1413, COST Action CA15219 DNAqua-Net STSM39774, and by the statutory funds of the University of Lodz. Data from Norway were generated in collaboration with the Norwegian Barcode of Life Network (NorBOL) funded by the Research Council of Norway (226134/F50) and the Norwegian Biodiversity Information Centre (70184209) where of the majority of the data were assembled through the project "Akvatiske insekter i Midt-Norge" and the project "Insekter i fuktige habitat i

Finnmark" funded by the Norwegian Taxonomy Initiative. This paper is also a result of the European Cooperation in Science and Technology (COST) Action DNAqua-Net (CA15219). There was no additional external funding received for this study. The funders had no role in study design, data collection and analysis, decision to publish, or preparation of the manuscript.

### Grant Disclosures

The following grant information was disclosed by the authors:
German Federal Ministry of Education and Research: FKZ 01LI1101 and 01LI1501. Bavarian State Ministry of Education and Culture, Science and the Arts. German Federal Ministry of Education and Research: GBOL2: BMBF #01LI1101B. Austrian Federal Ministry of Science, Research and Economy. European Agricultural Fund for Rural Development 2014-2020. Polish NCN: 2018/31/B/NZ8/03103. Klaudia Nowak. ITC Conference Grants: CA15219-1413, CA15219 DNAqua-Net STSM39774. University of Lodz. Research Council of Norway: 226134/F50. Norwegian Biodiversity Information Centre: 70184209. Norwegian Taxonomy Initiative. European Cooperation in Science and Technology (COST) Action DNAqua-Net (CA15219).

### Competing Interests

Jérôme Morinière is founder & CEO of AIM Advanced Identification Methods GmbH. All authors declare that they have no competing interests.

### Author Contributions

- Matthias Geiger conceived and designed the experiments, performed the experiments, analyzed the data, prepared figures and/or tables, authored or reviewed drafts of the paper, and approved the final draft.
- Stephan Koblmüller conceived and designed the experiments, performed the experiments, analyzed the data, prepared figures and/or tables, authored or reviewed drafts of the paper, and approved the final draft.
- Giacomo Assandri analyzed the data, authored or reviewed drafts of the paper, and approved the final draft.
- Andreas Chovanec analyzed the data, authored or reviewed drafts of the paper, and approved the final draft.
- Torbjørn Ekrem conceived and designed the experiments, analyzed the data, authored or reviewed drafts of the paper, and approved the final draft.
- Iris Fischer analyzed the data, authored or reviewed drafts of the paper, and approved the final draft.
- Andrea Galimberti analyzed the data, authored or reviewed drafts of the paper, and approved the final draft.
- Michał Grabowski analyzed the data, authored or reviewed drafts of the paper, and approved the final draft.
- Elisabeth Haring analyzed the data, authored or reviewed drafts of the paper, and approved the final draft.

- Axel Hausmann conceived and designed the experiments, analyzed the data, authored or reviewed drafts of the paper, and approved the final draft.
- Lars Hendrich analyzed the data, authored or reviewed drafts of the paper, and approved the final draft.
- Stefan Koch analyzed the data, authored or reviewed drafts of the paper, and approved the final draft.
- Tomasz Mamos analyzed the data, authored or reviewed drafts of the paper, and approved the final draft.
- Udo Rothe analyzed the data, authored or reviewed drafts of the paper, and approved the final draft.
- Björn Rulik analyzed the data, authored or reviewed drafts of the paper, and approved the final draft.
- Tomasz Rewicz analyzed the data, authored or reviewed drafts of the paper, and approved the final draft.
- Marcia Sittenthaler analyzed the data, authored or reviewed drafts of the paper, and approved the final draft.
- Elisabeth Stur analyzed the data, authored or reviewed drafts of the paper, and approved the final draft.
- Grzegorz Tończyk analyzed the data, authored or reviewed drafts of the paper, and approved the final draft.
- Lukas Zangl analyzed the data, authored or reviewed drafts of the paper, and approved the final draft.
- Jerome Moriniere conceived and designed the experiments, performed the experiments, analyzed the data, prepared figures and/or tables, authored or reviewed drafts of the paper, and approved the final draft.

**Field Study Permissions**

The following information was supplied relating to field study approvals (i.e., approving body and any reference numbers):

For samples from Germany field work permits were issued by the responsible state environmental offices in Bavaria [Bayerisches Staatsministerium für Umwelt und Gesundheit, for the project: "Barcoding Fauna Bavarica"] and from the Amt für Natur- und Landschaftsschutz, Rhein-Sieg-Kreis (67.1–1.03–19/2016KRO).

Italian specimens were collected in part in protected areas and some of the collected species are included in the EU Habitats Directive. The Italian Ministry of the Environment, Land and Sea released a national permit for the collection of species included in European and Italian conservation directives or to collect samples in regional or national protected areas (Prot. n° 0031783.20-11-2019).

Austrian specimens were collected with permits from the provincial governments of Burgenland (A4/NN.AB-10097-5-2017 and A4/NN.AB-10200-5-2019), Lower Austria (RU5-BE-1489/001-208; RU5-BE-64/018-2018), Styria (ABT13-53S-7/1996-156 and ABT13-53W-50/2018-2) and Vienna (MA22-169437/2017).

Polish specimens from Wigierski National Park were collected under the permit no. 12/2018 issued by the Park authorities to Grzegorz Tończyk. All the other material did not require additional permits for legal collection.

None of the collected specimens from Norway were from areas where sampling is restricted. Thus, sample permits were not required.

## Data Availability

DNA barcode sequences, PCR primers, and trace files are available in the "DS-ODOGER - DNA barcode references for Central and Northern European Odonata" dataset at the BOLD dataset: DOI 10.5883/DS-ODOGER

GenBank accession numbers are available in the Supplemental Files.

## Supplemental Information

Supplemental information for this article can be found online at http://dx.doi.org/10.7717/peerj.11192#supplemental-information.

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
