# Peer review of "Coverage and quality of DNA barcode references for Central and Northern European Odonata"

_PeerJ, doi:10.7717/peerj.11192_

## Round 0.1 · original submission · Major Revisions

I have received now three comprehensive reviews about your manuscript. While all of them were very positive about your work, they also pointed out the need to correct or improve the text. Especially R1 and R2 provided several comments to which I agree on the need to reduce and simplify the discussion and also rearrange some parts of the Methods by bringing parts of the results.

Please, pay special attention to the comments of R2 about the introduction.

Reviewer 1 ·

Basic reporting

The manuscript is clearly written and appears to adhere to PeerJ policies.

Experimental design

The rationale and objectives of the study are clearly described and the Material and Methods section is well structured but more details should be included.

Validity of the findings

The Discussion section is critical and comprehensive, but the Conclusion section is long and extensive.

Additional comments

The authors present an interesting manuscript about the capacity to use DNA barcoding to identify dragonflies and damselflies (Insecta: Odonata) in Central and Northern Europe. The strength of this study is in the use and acquisition of genetic information from odonates of this region.

The manuscript has a good introduction despite small errors (e.g. Line 65 and 69). The methodology is clear, however, I think more detail would assist with the methods section. It is also necessary to identify the authors’ of the species description for practically all species. I suggest to move part of the last paragraph from Result section (L 337-343) to the Material and Methods section. Tables and figures are good, but it is necessary some adjustments. The discussion section is critical and comprehensive; and the references are appropriate in number and up-to-date.

Considering the above-mentioned issues, it is an important study, especially because of its originality, and should be considered for publication in PeerJ once the corrections and modifications have been carried out properly.

Point-to-point revision

Line 65. Please, add some reference here or update this value.
Line 69. Please, add some reference here.
Line 81. See Kalkman, V.J., Boudot, J., Bernard, R. et al. Diversity and conservation of European dragonflies and damselflies (Odonata). Hydrobiologia 811, 269–282 (2018). https://doi.org/10.1007/s10750-017-3495-6
Line 102. Update the information about it – “June 12th, 2019”
Line 203. Get the DOI number before the manuscript has been accepted. Also, you must show the GenBank number for each sequence in Suppl. File 1.
Line 303. What are you so sure about the misidentifications of specimens?
Lines 337-343. Move to the Methods section.
Line 339. Koroiva & Kvist, 2018
Line 516. See Blow et al.2020 (https://doi.org/10.1101/2020.06.06.137828)
Line 561. The conclusion section is long and extensive. Try to summarize this in a few sentences.
Line 843. Advances in Ecological Research, Volume 58, 2018, Pages 63-99
Figures 4 and 5. Please, remove “1a”, “1b”, “Plate IX”, etc. In addition, there is no prior information in the text about “ML-based estimation using model MKY+F+I” or “TIM2+F+G4”. Add this information to the Material and Methods section.
Tables 1 and 2. “* as of 2020-05-18 in BOLD including all records ….” – Explain this information in the Material and Methods section.
Supp. File 1. Please standardize the terms "Imago", "Adult", "A", "a", "I", "larva", etc.
Supp. Files 1 and 2 and References. The scientific names of species are italicized.

Reviewer 2 ·

Basic reporting

The manuscript is well written, the authors use clear langage throughout the text.
The introduction and background provided by the authors is complete and well referenced. However, some parts of the dicussion could be rewritten or even removed, making the whole section a bit shorter and easier to read. For example, the authors provide some background on Red List assesmentsand conservation issues for odonates. While this information is indeed interesting, I do not see the point of including it here, as the authors do not dicuss any of these points later on. I suggest the authors should delete for example the paragraph on conservation (lines 82-86).
Lines 100-102: I suggest to expand a bit more on this background info (e.g. by providing absolute numbers, or details on particular species that are missing from the BOLD database and that are relevant), as it provides the authors the starting point for their work, and hence they need to clearly state how their work contributes to fill this knowledge gap.
Line 108: the authors mention that «A wider geographic coverage is lacking..» Wider as compared to what? Perhaps they should briefly state the current gepgraphical coverage for odonates in the databases.
Line 145: The authors may provide the readers with references to cryptic odonate species that have been described, for example;
Mitchell A, Samways MJ (2005) The morphological ‘forms’ of Palpopleura lucia (Drury) are separate species as evidenced by DNA sequencing (Anisoptera : Libellulidae). Odonatologica, 34, 173– 178.
Damm S, Schierwater B, Hadrys H (2010) An integrative approach to species discovery: from character‐based DNA barcoding to ecology. Molecular Ecology, 19, 3881– 3893.
Vega-Sanchez et al (2020) Hetaerina calverti (Odonata: Zygoptera: Calopterygidae) sp. nov., a new cryptic species of the American Rubyspot complex Zootaxa 4766 (3)
Overall, I think that the whole introduction section would benefit from a slight re-focus on a few main points, such as the following:
1. Background on odonates as important in monitoring freshwater ecosystems.
2. A good coverage of odonate species in the BOLD database DOES EXIST, BUT THERE IS A GAP IN (and here state that there are severa instances of species represented by a single individual, and also several species (and why are these relevant?) missing from the database).
3. What did the authors do to fill these gaps?
FIGURES AND TABLES:
Figure 1 could be changed slightly by colouring the circles (besides the differences in size), as in some cases it is not easy to distinguish their sizes.
In Figures 2 and 3, please indicate in the legend which are the clades in which you identified BIN discordances. I understand that these appear in blue in both cases, but still this should be indicated in the figure legends.
There are some instances of references cited in the main text which do not appear in the reference list(or they do but the publication years do not match):
Line 66 – Kalkman et al. 2008
Line 93 – Leese et al 2018
Line 141 – Schlick-Steiner et al. 2014
Line 148 – Macher et al. 2016
Line 397 – Hawlitschek et al. 2016
OTHER MINOR COMMENTS:
Line 56 – Cordulia aenea should be in italics.
Line 530 – Please correct Cordero-Rivera, also in the reference list.

Experimental design

The authors should state in a more clear way how their study fills a knowledge gap (see my previous comments). It would be good that the authors explained the reason why different primer combinations and/or DNA extraction protocols are used. I know that this might be due to simple logistical reasons (each laboratory has usually standarized work procedures) but still it would be good to clarify this.
The methodology regarding the laboratory methods used to obtain the barcode sequences is not explained in a clear way. I acknowledge it is difficult to put together different methods used in different institutions/laboratories; however this particular section of the manuscript as it is written now, makes it a bit difficult for the reader to follow the description and hence to clearly understand the methodology used by the authors. For example:
In lines 162-167, the authors enumerate the different institutions that collaborated with the study by providing specimens for DNA extraction. Would be good that they clarify what do the acronyms used stand for (e.g. SNSB-ZSM, NHMW or ZFMK).
Line 164, the authors mention here the University of Lodz in Poland; however, later on (line 208) they refer to the DNAqua-Net from Poland. Please clarify this.
Line 186, «Most studied specimens were adults (see below) of which the majority were stored in >96%» What about the others? Where they dry-preserved specimens? Were they preserved in different ethanol concentration? Please clarify.
Lines 207-248, It is not really clear where the procedures (DNA extraction, PCR and sequencing) were carried out for each of the samples contributed by each country/institution involved. For example, in this section the authors mention the CCDB as a place where some samples were sequenced, but then in the results section (line 277) they mention GBOL (?). The authors should add some statement at the beginigng of this section in which they explain where all the samples from the contributing institutions mentioned previously (lines 162-167) were processed.
There is no mention to the starting tissue for DNA extraction. Was it legs? Thoracic muscle?
Line 220, why does the CCDB uses a different primer set to amplify the samples from DNA DNAqua-Net from Poland?
Line 224, where were samples from ZFMK processed? Was it also at CCDB?
Line 226, what does KFUG stands for? This is mentioned for the first time here.
Line 231, is the NHMW located at the University of Graz, or these are two different insititutions from Austria?
Lines 234-236, Reference for primers is missing.
Lines 237-238, Reference for primers is missing.
Lines 245-246, Reference for primers is missing.
Overall, this whole section (Laboratory Methods) needs to be written in a more clear and concise way. For the sake of clarity and readability, perhaps the authors should consider providing detailed primer information in a separate table, including the institution/laboratory that used each primer pairs, annealing temperature, and original references for the primer pairs; and/or any other information that might be useful for replication of the experiments.
Results section, line 280, perhaps the authors could further explain the reason for those 3 individuals not being assigned to a BIN. Was it because of low-quality sequence data? Presence of nuclear copies? Contamination issues?

Validity of the findings

The discussion section is far too long. Also, the conclusions that the authors make regarding those instances of BIN discordance or BIN sharing go in my oppinion far beyond to what is supported by their results. While I acknowledge that some speculation and hypothesis delineating is needed in a manuscript discussion, I think that in this particular case there are instances in which the authors make conclusions that go far beyond the resolution power of the data available to them (only one nuclear marker), which also makes the reading of this section a bit difficult. I suggest that the authors should cut out this section, focusing on their main findings, discuss briefly potential explanations for their results and mention future avenues of research, which are basically the same for all examples presented: Increase the geographical/specimen sampling, and increase the number and type of markers used (nuclear and mitochondrial). I detail below some specific issues:
Lines 346-355, I think this paragraph is more suited to be provided as background. The Discussion section should open by stating the main findings of the work (perhaps something like current lines 371-376).
Lines 409-410, perhaps the authors could also mention somwhere in the discussion that COI might not be the best mitochondrial marker for resolving species-level relationships? (see e.g. Chen et al (2018) Systematic Entomology; DOI: 10.1111/syen.1229 or De Mandal et al (2014) DNA Barcodes; DOI: 10.2478/dna-2014-0001)
Lines 413-414, This statement seems to be repetitive as it says basically the same as lines 400-401.
The text from lines 422 to 559 needs to be shortened, focusing only on discussing the results in light of the current data available - I feel there is a bit too much speculation in some cases. Also, the authors should present their arguments and also the background they provide on the differens species in question, in a more clear, ordered way.
As a note for potential explanations regarding haplotype sharing between different species, the authors should also consider the possible presence of maternally inherited endosymbionts. There has been recent progress in screening odonates for endosymbionts such as Wolbachia or Rickettsia. I think this might be relevant here, as for example in the case of Coenagrion, Thongprem et al have found that some species of this genus from the UK are infected with Rickettsia.The authors should mention this as a potential explanation for some of their findings, along with the ones that they already provide:
Thongprem et al 2020 Incidence and Diversity of Torix Rickettsia–Odonata Symbioses Microbial Ecology https://doi.org/10.1007/s00248-020-01568-9
Lorenzo-Carballa et al 2019 Widespread Wolbachia infection in an insular radiation of damselflies (Odonata, Coenagrionidae) Scientific Reports https://doi.org/10.1038/s41598-019-47954-3
Salunkhe, R. C. et al. Distribution and molecular characterization of Wolbachia endosymbionts in Odonata (Insecta) from Central India by multigene approach. Current Science. 108(5), 971–978 (2015).
Lines 450-451, the argument on hybridization between Anax species sounds a bit too speculative. I suggest removing lines 450-457 and focus only on what you can argue with your data (currently lines 457-464).
Lines 492-497, too speculative. I suggest removing this bit.
Lines 541, The authors need to be careful to the arguments they provide. For example when mentioning «Interestingly, though, this second C. aenea clade groups with COI data of C. amurensis from Japan (Fig. 5)», they should be aware that they only have one sequence of C. amurensis available, and that the whole picture could change if they sequenced more individuals. Again, the authors need to remove any speculative sentence/paragraph. The same goes for the closing argument of this section «Our finding that the barcodes of some European C. aenea cluster with C. amurensis, suggests introgression of C. amurensis into C. aenea and a rapid spread of this foreign haplotype across large parts of the species’ distribution range».

Additional comments

This manuscript by Geiger et al reports the results of a barcode analysis of nearly all European Odonate species. The aim of this work was to fill the gap that currently exists in the BOLD database, in which, despite having covered nearly all European dragonflies(Weigand et al 2019); several instances exist of odonate species that are represented by a single individual in the BOLD database, and even some species missing from this database.
To fill these gaps, Geiger et al compiled an extensive dataset of nearly 700 specimens of odonates (including adults and larvae), which were sequenced for the COI barcode fragment. The authors examined their dataset using the BOLD database and metrics included there, and perfomed also ML analysis.
They found that nearly 90% of the species included in the study could be readily identified by using the COI barcoding sequence. Interestingly, they found instances of shared barcodes between morpholgically distinct species, as well as deep divergence among representatives of the same morphological species. The authors discuss their results, focusing on the need for more in depth analyses of these discordances.
This work makes a good contribution to the barcode database for European odonates, which is a crucial and neccesary step prior to the use of DNA-based identification tools (as the authors clearly state throughout their manuscript); and as such, I think that the article deserves publication in PeerJ. However, there are some issues related to the methodology description, as well as to the conclusions that the authors make in light of their results; which need to be dealt with prior to manuscript acceptance.
I have tried to address some of these issues, and attach also a copy of the manuscript main document with several other minor comments. I do hope that my review helps to improve this manuscript, and I look forward to receive a new version of the manuscript for reviewing and, eventually, publication in PeerJ.

Annotated reviews are not available for download in order to protect the identity of reviewers who chose to remain anonymous.

·

Basic reporting

The article is on the whole clear and well written. There are a small number of ambiguous statements, likely due to English not being a first language and these should be corrected, e.g. lines 68 ("...people affectionate by nature" - the meaning intended is surely people who like nature. Perhaps "natural historians"?).

A detailed background is presented and the work is well referenced, well structured, and provided with appropriate figures and tables.

Experimental design

No comment.

Validity of the findings

The underlying data has been provided, although it would be beneficial to provide a DOI for the dataset on BOLD. Conclusions are concisely stated and do not overreach the results.

For Figs. 2 and 3, the notation in parentheses after species names (e.g. (NO 4, DE 6, PL 7) should be explained in the legend to each figure.

Some enigmatic results in Table 2 (e.g. BIN BOLD:AAA2218 contains approx. 15 species of Enallagma) should be discussed in the text. At the very least there should be a comment in the appropriate column of Table 2 for this BIN.

---

## Round 0.2 · accepted · Accept

I have received one final review of your work and I'm glad to recommend it for publication as it is. Thank you for your contribution.

Reviewer 2 ·

Basic reporting

I am happy to see the revised version of the manuscript by Geiger et al on Odonata barcoding, in whiche the authors have addressed all the concerns by the reviewers. This new version of the manuscript has greatly improved compared to the previous one, and I believe it is now ready for publication in PeerJ.

Experimental design

The methods are now clearer and written in a way less prone to confussion for the readers, the reading of the whole mats & methods section is now much easier.

Validity of the findings

no comment

Additional comments

I am really happy to see this manuscript in its current form, and it will be great to see it published soon in PeerJ.